

# Surgical antibiotic prophylaxis use and infection prevalence in non-cosmetic breast surgery procedures at a tertiary hospital in Western Australia—a retrospective study

Ainslie Lavers[1], Wai Siong Yip[1], Bruce Sunderland[1], Richard Parsons[1], Sarah Mackenzie[2], Jason Seet[2] and Petra Czarniak[1]

[1] School of Pharmacy and Biomedical Sciences, Curtin University, Perth, Western Australia, Australia
[2] Pharmacy Department, Sir Charles Gairdner Hospital, Perth, Western Australia, Australia

## ABSTRACT

**Background.** Surgical site infections (SSIs) are a common complication following breast surgery procedures, despite being considered a clean surgery. The prevalence of SSIs can be minimised with the appropriate use of antibiotic prophylaxis as outlined in the Australian Therapeutic Guidelines (eTG). The aims of this study were to evaluate adherence to the eTG for antibiotic prophylaxis in breast surgery procedures at a Western Australian teaching hospital following an update of the guidelines in 2014 and examine the impact of prophylactic antibiotics on SSI incidence and length of hospital stay.

**Method.** A retrospective cross-sectional study which reviewed medical records from a random sample of 250 patients selected from 973 patients who underwent breast surgical procedures between February 2015 and March 2017.

**Results.** Overall adherence to current eTG occurred in 49.2% (123/250) of operations. Pre-operative and post-operative antibiotics were prescribed in 98.4% (246/250) and 11.2% (28/250) operations respectively. Adherence rates to three specific elements of the eTG (drug prescribed, drug dosage and timing of administration) were 91.6% (229/250), 53.6% (134/250) and 86.4% (216/250) respectively. For the 14.4% (36/250) patients with relevant drug allergies, there was zero adherence to the eTG. Overall recorded SSI prevalence was low at 5.2% (13/250). The mean length of stay in patients (2.3 ± 1.7 days) was not influenced by level of eTG adherence ($p = 0.131$) or SSIs ($p = 0.306$).

**Conclusion.** These data demonstrate a significant improvement in overall adherence to the eTG from 13.3% to 49.2% ($p = < 0.001$). The level of detected SSIs in this study was low. Further improvement is necessary with respect to prescribing appropriate antibiotic dosages and for those with allergies.

Corresponding author
Petra Czarniak,
P.Czarniak@curtin.edu.au

## INTRODUCTION

Surgical site infections (SSIs) are a common complication following breast surgery procedures, despite being considered a 'clean surgery' (*Cabaluna et al., 2013*; *Craft, Damjanovic & Colwell, 2012*). SSIs are also the second most common adverse event in breast surgical patients with incidences of infection typically ranging from 0.8–26% (*Cabaluna et al., 2013*; *Craft, Damjanovic & Colwell, 2012*; *Gulluoglu et al., 2013*).

The prevalence of SSIs can be minimized with the appropriate use of pre-operative antibiotic prophylaxis (*Ariyan et al., 2015*; *Jones, Bunn & Bell-Syer, 2014*). This has been demonstrated in patients undergoing breast surgery (*Jaber et al., 2017*). A recent study by *Jaber et al. (2017)* in 2014 evaluated the appropriateness of surgical antibiotic prophylaxis for breast surgery procedures and found a statistically significant relationship between pre-operative prophylactic antibiotic use and successful SSI prevention. The burden of SSIs not only includes an impact on patient recovery, the associated cost of hospital readmission and subsequent adjuvant treatment, but also SSI-related patient morbidity and mortality. Hence the application and adherence to evidence-based guidelines should be considered to minimize rates of SSIs (*Cabaluna et al., 2013*; *Crawford et al., 2016*; *Manian, 2014*).

In Australia, prescribers are guided by the Australian Antibiotic Therapeutic Guidelines (eTG) which aim to promote the quality use of medicines by publishing evidence based standard protocols and up-to-date therapeutic information. In 2014 updates were made to the eTG and a new section on surgical prophylaxis of breast surgery was included in the guidelines. According to the current eTG, patients undergoing breast surgery should receive pre-operative cefazolin 2 grams intravenously (IV) within 60 min, ideally 15 to 30 min before surgical incision. For patients with immediate hypersensitivity to penicillin, vancomycin 15mg/kg IV, started 30 to 120 min before surgical incision at a rate 10mg/min is recommended (*Antibiotic Expert Groups, 2014*). Vancomycin should be added to cefazolin in suspected patients or patients infected with methicillin-resistant *Staphylococcus aureus* (MRSA) (*Antibiotic Expert Groups, 2014*). Post-operative antibiotics have not been indicated in these guidelines however the guidelines mention the potential benefits in obese patients, patients being treated with radiation therapy or in breast reconstruction patients (*Gulluoglu et al., 2013*; *Manian, 2014*; *Antibiotic Expert Groups, 2010*; *Townley et al., 2015*; *Antibiotic Expert Groups, 2014*; *Huang et al., 2015*; *Phillips et al., 2013*; *Viola, Raad & Rolston, 2014*).

Prior to the update, prescribers would follow the protocol for 'Head, Neck and Thoracic' procedures (*Antibiotic Expert Groups, 2010*). The 2014 study conducted by Jaber et al., evaluated prescriber adherence to the eTG in breast surgeries performed at a single Western Australian teaching hospital prior to the guidelines update in 2014 (*Jaber et al., 2017*). Researchers reported low adherence to guidelines (13.3%) and proposed an improvement would be seen after the update (*Jaber et al., 2017*). Since that time no study has been performed to evaluate the effectiveness of these updates nor an improvement in adherence.

## Aim of the study

The aims of this study were to evaluate adherence to the eTG for antibiotic prophylaxis in breast surgery procedures at a Western Australian teaching hospital since the 2014 eTG update and to examine the impact of prophylactic antibiotics on SSI incidence and length of hospital stay.

## Ethical approval

The study was approved by the Sir Charles Gairdner Group Human Ethics and Research Committee (QI:14155) as a quality improvement activity to enhance the safety of medicines use. Approval of this piece of research included the option to publish the data with a waiver of individual patient consent. Ethical approval was also granted by the Human Research Ethics Committee at Curtin University, Perth, Western Australia (HRE2017-0189).

## METHODS

This retrospective cross-sectional study was conducted at Sir Charles Gairdner Hospital (SCGH) in Perth, Western Australia. A total of 973 patients who underwent a breast surgery procedure between February 2015 and March 2017 at SCGH were identified by the Pharmacy Department at SCGH using Medicare breast surgery codes by means of an electronic database. The search involved identifying patients with International Classification of Disease (ICD-10) codes. C50 was used as well as Australian Medicare Benefits Schedule principle procedures codes 31,500 to 31,566 which formed a list of patients not ordered by date, code or medical record number. Patients with missing important information such as patient treatment, detailed antibiotic use and type of surgery were excluded. In Australia, Medicare is a Commonwealth government universal health care system that provides citizens with access to many health services at minimal cost, including free treatment in public hospitals. The Medicare Benefits Schedule is a listing of the Medicare services subsidized by the Australian government.

A sample of 250 patients was selected using an online randomiser (*Townley et al., 2015*; *Social Psychology Network, 2017*). This sample size was selected so that the 95% confidence for the prevalence of guideline adherence would be no wider than ±6%. This degree of precision was considered adequate for this study. Only the first operation for each patient was included for analysis. For patients whose files were unavailable, the next patient sequentially from the random generator list was used.

De-identification of patient and prescriber details was performed during data collection to ensure confidentiality throughout the study. For this study details gathered from the medical records were entered into a password protected Microsoft Excel® Spreadsheet. Data extracted included patient demographics (age, gender, weight, height and allergies), the code of the surgeon who performed the procedure, the patients' ward, the procedure type, surgery time, any pre-operative or post-operative antibiotics given (drug name, dose, route of administration, time of administration and frequency of antibiotics given), adverse drug reactions for both pre- and post-operative antibiotics, signs of infection and the patient's length of stay in the hospital.

**Table 1 Patient demographics.**

| Demographic | Gender | Mean (± standard deviation) |
|---|---|---|
| **Age** (years) | **Female** ($n = 247$) | $56.8 \pm 13.7$ |
| | **Male** ($n = 3$) | $46.7 \pm 22.9$ |
| **Weight** (kg) | **Female** ($n = 246$) | $76.1 \pm 20.4$ |
| | **Male** ($n = 3$) | $92.7 \pm 10.8$ |
| **Height** (cm) | **Female** ($n = 235$) | $162.4 \pm 8.0$ |
| | **Male** ($n = 3$) | $183 \pm 1.7$ |

In order to calculate adherence to the current version of the eTG, four characteristics of pre-operative antibiotics recorded for each patient (antibiotic drug, dose, route of administration and timing of administration) were assessed against the current guidelines, which recommend the use of cefazolin 2 grams (IV) within 60 min, ideally 15 to 30 min before surgical incision (*Antibiotic Expert Groups, 2014*). To be classified as 'adherent' the prescriber must have satisfied all of these criteria of the eTG; if not, the treatment was rendered as 'non-adherent'. For all other determinations and statistical analysis in this study the Statistical Package for the Social Science (SPSS) version 23 software was utilised. The statistical significance of univariate associations between SSIs and adherence, procedure, prescriber, length of hospital stay (0–2 days vs 3 days or more), and the association between adherence and length of hospital stay were assessed using the Chi-square test or Fisher's Exact test, as appropriate. A Chi-square test was also used to compare levels of adherence to the eTG before and after the 2014 eTG update. A logistic regression model was used to identify any multivariate associations with the dependent variables (guideline adherence, or development of a SSI). Results for the logistic regression are provided as the odds ratio, its 95% confidence interval (95% CI), and *p*-value. A *p*-value $<0.05$ was taken to indicate a statistically significant association in all tests.

## RESULTS

A total of 250 patients were analysed in this study. Most, 98.8% (247/250), were females (Table 1). The 14.4% (36/250) of patients who had a documented allergy to either penicillin or cefalosporins were all female.

There were 25 different types of breast procedures recorded (Table 2). Many patients underwent multiple procedures in the one operation with an average of 2.1 procedures/operation recorded per patient (median: 2 procedures/patient; range 1–4). A total of 521 procedures were performed during the 250 operations. The most common procedures performed were sentinel node biopsy ($n = 120$; 23.0%) and wide local excision ($n = 88$; 16.9%). All male patients received a unilateral mastectomy with one male patient also receiving a sentinel node biopsy. There was no statistically significant association between any particular operation and development of a SSI.

There was missing information on specific adherence elements for 10.4% (26/250) of patients, which related to timing of antibiotic administration. These patients were classified as non-compliant with respect to timing. Management of 123 patients (49.2%) was found

**Table 2** Types of procedures and association with surgical site infections (SSI). *P*-values were obtained from Fisher's Exact test, unless otherwise specified.

| Type of procedure | Frequency of procedure | Frequency (%) of SSI | *p* |
|---|---|---|---|
| Axillary node clearance | 52 | 1 (1.9%) | 0.313 |
| Mastectomy (unilateral) | 84 | 4 (4.8%) | 1.0 |
| Mastectomy (bilateral) | 12 | 2 (16.7%) | 0.123 |
| Sentinel node biopsy | 120 | 5 (4.2%) | 0.480[*] |
| Wide local excision | 88 | 3 (3.4%) | 0.552 |
| Hook wire local excision | 68 | 1 (1.5%) | 0.196 |
| Excision | 27 | 2 (7.4%) | 0.638 |
| Reconstruction/expanders | 6 | 0 (0.0%) | 1.0 |
| Implant insertion | 1 | 0 (0.0%) | 1.0 |
| Microdochectomy | 6 | 0 (0.0%) | 1.0 |
| Duct excision | 5 | 0 (0.0%) | 1.0 |
| Lumpectomy | 2 | 0 (0.0%) | 1.0 |
| Seed removal | 1 | 0 (0.0%) | 1.0 |
| Soft tissue biopsy | 3 | 1 (33.3%) | 0.149 |
| Abscess drainage | 5 | 1 (20.0%) | 0.236 |
| Abscess incision | 5 | 1 (20.0%) | 0.236 |
| Breast reduction | 1 | 0 (0.0%) | 1.0 |
| Hematoma drainage | 1 | 1 (100%) | 0.052 |
| DIEP flap breast reconstruction | 6 | 1 (16.7%) | 0.277 |
| Wound exploration | 2 | 0 (0.0%) | 1.0 |
| Seed localization | 2 | 0 (0.0%) | 1.0 |
| Excision biopsy | 20 | 0 (0.0%) | 0.608 |
| Re-excision | 2 | 0 (0.0%) | 1.0 |
| Lipofilling | 1 | 0 (0.0%) | 1.0 |
| Liposuction | 1 | 0 (0.0%) | 1.0 |

**Notes.**

[*] *p*-value obtained from the Chi-square statistic.

to be compliant with the eTG (Table 3). The 95% CI for overall compliance was found to be: 43.0%–55.4%. Adherence to specific factors of the eTG requirements were: correct drug 91.6% (229/250), correct dose 53.6% (134/250), correct route of administration 97.2% (243/250) and correct timing of administration 86.4% (216/250) (Table 3). There was zero adherence to the eTG regarding patient allergies, although the allergies were not recorded as immediate hypersensitivity reactions to penicillin. Despite 37 patients reported as having an allergy to penicillin or cefalosporin, 67.6% (25/37) received a cefalosporin (cefalozolin), 29.7% (11/37) received clindamycin, and only one of these patients (2.7%) did not receive an antibiotic.

Adherence to the eTG was found to be significantly improved since the study by Jaber et al. reported 13.3% compliance ($p < 0.001$, Chi-square test). In the current study, 5.2% (13/250) of patients developed a reported SSI after surgery. No statistically significant relationship was found between adherence to the eTG and SSIs (OR ratio: 1.36; 95% CI [0.43–4.30]; $p = 0.597$). Of the patients who had received appropriate pre-operative
**Table 3** Levels of adherence to specific elements of the Australian Antibiotic Therapeutic Guidelines in 250 operations (blue colour indicates element complied with; blank colour indicates element not complied with).

| Correct adherence to prescribing parameters when compared to the *Therapeutic Guidelines: Antibiotic* Version 15 | | | | Number of operations | |
|---|---|---|---|---|---|
| Antibiotic selection | Route | Dose | Timing | (n = 250) | (%) |
| | | | | 123 | 49.2 |
| | | | | 79 | 31.6 |
| | | | | 11 | 4.4 |
| | | | | 13 | 5.2 |
| | | | | 2 | 0.8 |
| | | | | 1 | 0.4 |
| | | | | 12 | 4.8 |
| | | | | 5 | 2.0 |
| | | | | 1 | 0.4 |
| | | | | 3 | 1.2 |

antibiotic treatment, 6.5% (8/123) developed a reported SSI, while 4.9% (5/103) of the patients who did not receive appropriate treatment developed an SSI.

On average patients stayed at the hospital for a mean (±standard deviation) of 2.3 ±1.7 days (median: 2 days; range 1–16 days). With this length of stay divided into short (0–2 days) vs longer (3 or more days), no significant association was found between level of adherence and length of stay (OR: 2.0; 95% CI [0.5–7.6]; $p = 0.3144$) nor SSI occurrence ($p = 0.596$). A significant relationship was found between using cefazolin and a decreased length of stay ($p = 0.037$), with 87.8% (201/229) of those who were given cefazolin staying 1-2 days, compared to 71.4% (15/21) of those who were not taking cefazolin. There were 4.6% (10/216) patients with a length of stay less than three days who developed a SSI, whilst 8.8% (3/34) of those who stayed longer than two days developed a SSI.

A total of seven surgeons were recorded in this study. There was no statistically significant relationship found between prescribers and level of adherence ($p = 0.631$) with prescriber adherence to the eTG ranging from 25 to 100%. Similarly, between prescribers and developing an SSI ($p = 0.748$) with individual SSIs rates for prescribers ranging from 0-14.3%. No significance was found between prescribers and whether they gave pre-operative antibiotics with the rates of pre-operative antibiotic given to patients for each prescriber ranging between 90.7% and 100%. A statistically significant association was seen between prescribers and whether post-operative antibiotics were given ($p<0.05$) with 11.2% (28/250) in patients having received post-operative antibiotics despite not being specified by the eTG. A majority of patients who received post-operative antibiotics (57.1%; 16/28) were prescribed cefalexin, however other antibiotics including clindamycin (14.3%; 4/28), dicloxacillin (10.7%; 3/28), cefazolin (3.6%; 1/28), flucloxacillin (3.6%; 1/28), amoxicillin (7.1%; 2/28) amoxicillin with clavulanic acid (3.6%; 1/28) were also prescribed.

## DISCUSSION

This is the first study conducted since the introduction of the new eTG in 2014. The adherence level of 49.2% (123 operations) was higher ($p < 0.001$) than the 13.3% (20 operations) reported by *Jaber et al. (2017)* and zero adherence by *Habak et al. (2013)* who investigated adherence to the previous version of the eTG. This is an improved result compared to the previous study yet still not reflective of strict adherence to the breast surgical antibiotic prophylaxis eTG guidelines. The main factor that contributed to non-adherence, was antibiotic dosage which accounted for 46.4% of operations. This was different to the study conducted by *Jaber et al. (2017)* where inappropriate timing of the antibiotic was the dominant factor contributing to non-adherence. To ensure adherence to guidelines, there are measures that can be implemented as suggested by *Nabor, Buckley & Lapitan (2015)*. These measures include medical education presentations and laminated summaries of guidelines posted in surgical areas such as operating rooms (*Nabor, Buckley & Lapitan, 2015*).

Antibiotic prophylaxis was used in 98.4% of patients who underwent a breast procedure which was higher than the 92.7% and 53% reported by *Jaber et al. (2017)* and *Habak et al. (2013)* respectively. Jaber et al. investigated the use of prophylactic antibiotics in 150 breast operations in 150 patients whilst Habak et al. investigated prophylactic antibiotic use in 134 breast operations in 95 patients. Both studies were carried out in Western Australian hospitals (*Jaber et al., 2017*; *Habak et al., 2013*). This shows an increased use of prophylactic antibiotics in the last two evaluations.

This study related to the oncological management of breast surgery as part of treatment. Despite the inclusion of one patient who had breast reduction and one patient who had implant insertion, these surgeries were related to an original life-saving procedure rather than a cosmetic procedure, as these are not performed at the study hospital. Antibiotics play an important role in these types of procedures as reported by *Khan (2010)*.

Of the 250 patients analysed in this study, 37 patients were recorded with an allergy to penicillin or cephalosporin, although there was no evidence that these were associated with immediate hypersensitivity reactions. It was found that none of the prescribers adhered to the eTG and had prescribed either cefazolin or clindamycin instead of vancomycin as stipulated in the eTG. Vancomycin or clindamycin are second-line choices for SSI prophylaxis in cases where cefalosporin antibiotics were contraindicated (*Baghaki, Soybir & Soran, 2014*). Possible reasons for reluctance by clinician to prescribe vancomycin include adverse effects such as red man syndrome which may limit its use for some surgeons. Furthermore, vancomycin is contraindicated in patients with renal impairment (*Bratzler et al., 2013*).

No relationship was found between adherence to therapeutic guidelines and SSIs ($p = 0.596$). A recent study by Yang et al. also reported that there was no significant difference between SSIs and cefazolin administration in patients who underwent breast procedures such as a mastectomy, whereas development of SSIs with and without prophylactic pre-operative antibiotics was 7.2% and 4.2% respectively (*Yang et al., 2017*). In their paper they suggested that to decrease the development of post-operative SSIs other factors such

as hypertension, diabetes, and advanced age should be taken into consideration (*Yang et al., 2017*).

While no significant association was found between the different operations and occurrence of a SSI, hematoma drainage has been clinically shown to impart an increased rate of SSIs for patients especially with a longer drain duration (*Eroglu et al., 2014*). The study was able to collect data only on SSIs that were treated at the hospital. Other patients may have developed an SSI but were treated by a general practitioner or at another hospital.

There is a lack of consensus in current practice for the use of pre-operative antibiotic prophylaxis and this is largely attributed to the lack of trial evidence in preventing SSIs in general patients after breast surgery (*Ng et al., 2007*). Also, the variation in guidelines and adherence can be attributed to changing patient risk factors. For instance, in a study by *Eroglu et al. (2014)* researchers reported that prescribers were more inclined to prescribe antibiotics prophylactically in patients with risk factors which included older age, diabetes mellitus, immunodeficiency and those who underwent pre-operative chemotherapy or radiotherapy prior to breast surgical procedures. Other risk factors that may contribute to post-operative infections in breast surgery include length of surgery, type of surgery, smoking, steroid use, seroma, hematoma, surgical drain, second drain placed, prolonged close suction drainage and immediate breast reconstruction (*Vilar-Compte et al., 2004*; *Throckmorton et al., 2009*; *Tejirian, DiFronzo & Haigh, 2006*; *Penel et al., 2007*; *De Blacam et al., 2012*; *Vilar-Compte et al., 2009*; *Gao et al., 2010*; *Olsen et al., 2008*; *Xue et al., 2012*). When risk factors are present in patients undergoing breast surgery, the administration of prophylactic antibiotics should be taken into account. This is supported by a recent study by *Vieira et al. (2016)* which reported that SSIs were significantly more common in the control group that did not receive antibiotic prophylaxis despite having risk factors.

For patients that are not at risk for SSIs after breast surgery, routine antibiotic prophylaxis is not necessary (*Gulluoglu et al., 2013*; *Vilar-Compte et al., 2004*; *Tejirian, DiFronzo & Haigh, 2006*; *Olsen et al., 2008*). According to *Xue et al. (2012)* antibiotic prophylaxis is not an independent protective factor in SSI development and systematic administration for breast surgery is not necessary for general patients but may be considered if other risk factors were present.

In this present study no statistical significance was found between prescribers and adherence to the eTG ($p = 0.631$). As mentioned previously there is a lack of clinical studies which support the use of antibiotic prophylaxis which may have influenced prescribers in the present study. The lack of consensus of antibiotic prophylaxis has also been reported by breast surgeons in the United Kingdom, following nationwide surveys. (*Ng et al., 2007*) The eTG does not indicate that antibiotic prophylaxis is discretionary (*Antibiotic Expert Groups, 2014*).

Where patients were prescribed post-operative antibiotics, these did not adhere to the eTG in this study. Post-operative prophylaxis should not exceed 24 h and should be considered on an individual patient basis. (*Antibiotic Expert Groups, 2014*) Similar results have been reported by *Jaber et al. (2017)* and *Habak et al. (2013)*. According to other studies to prevent SSIs after breast and/or axillary surgery some surgeons prefer post-operative prophylaxis for patients with drains. In patients

receiving a surgical drain, mastectomy, immediate reconstruction or receiving prior radiation therapy or chemotherapy, pre- and post-operative prophylactic antibiotics are used (*Vilar-Compte et al., 2004*; *Throckmorton et al., 2009*; *Tejirian, DiFronzo & Haigh, 2006*; *De Blacam et al., 2012*; *Olsen et al., 2008*). *Phillips et al. (2013)*, *Manian (2014)* *Viola, Raad & Rolston (2014)*, and *Elbur et al. (2013)* each have emphasised the increased risk of antibiotic resistance and drug-related complications such as *Clostridium difficile* when post-operative antibiotic prophylaxis was used after breast surgeries, due to the lack of evidence.

It was also found that the mean length of stay was not influenced by the level of eTG adherence. However, the length of stay and SSIs were reduced when cefazolin was given. This somewhat supported the study conducted by *Toor et al. (2015)* who found that the administration of prophylactic antibiotics not only led to reduced SSIs but also led to a shortened hospital stay.

The strength of this study was a larger sample size ($n = 250$) compared to the previous study conducted by Jaber et al. which used a sample size ($n = 150$). The results should be generalisable to this hospital but may not reflect practice elsewhere.

Limitations of this study included the sample size ($n = 250$) which restricted the ability to identify large numbers of SSIs. Patients might seek treatment for SSIs from their general practitioners or other hospitals without returning to SCGH which may raise the incidence of SSIs recorded within this study cohort. A further limitation was that detailed examination of SSIs and the causative factors such as pathogens, appropriate hand washing and other procedures, were beyond the scope of this study. Also, missing information within 26 patient medical files regarding adherence, specifically the timing of antibiotic administration meant these patients were excluded from the overall compliance assessment.

## CONCLUSIONS

There was a significant improvement in adherence to surgical antibiotic prophylaxis guidelines in breast surgery from 13.3% to 49.2% compared to a similar study conducted by *Jaber et al. (2017)*. Further improvement is necessary especially with respect to recording of antibiotic timing of administration, adherence to guidelines for antibiotic dosage and also when allergy is reported to the primary recommended antibiotic, that the recommended alternative antibiotic is selected. From the available evidence, incidence of SSIs identified in the study was low and there was no relationship with adherence to guidelines, indicating that the prophylaxis followed was reasonably effective.

## ACKNOWLEDGEMENTS

The authors acknowledge the assistance of the medical records department of the research hospital for their support and assistance.

### Funding

The authors received no funding for this work.

### Competing Interests

The authors declare there are no competing interests.

### Author Contributions

- Ainslie Lavers and Wai Siong Yip performed the experiments, analyzed the data, prepared figures and/or tables, authored or reviewed drafts of the paper, approved the final draft.
- Bruce Sunderland and Richard Parsons conceived and designed the experiments, analyzed the data, contributed reagents/materials/analysis tools, authored or reviewed drafts of the paper, approved the final draft.
- Sarah Mackenzie and Jason Seet conceived and designed the experiments, analyzed the data, authored or reviewed drafts of the paper, approved the final draft, provided access to hospital medical records.
- Petra Czarniak conceived and designed the experiments, analyzed the data, contributed reagents/materials/analysis tools, prepared figures and/or tables, authored or reviewed drafts of the paper, approved the final draft.

### Human Ethics

The following information was supplied relating to ethical approvals (i.e., approving body and any reference numbers):

The study was conducted at Sir Charles Gairdner Hospital (SCGH) in Perth, Western Australia and approved by the Sir Charles Gairdner Group Human Ethics and Research Committee (QI:14155) as a quality improvement activity to enhance the safety of medicines use. Ethical approval was also granted by the Human Research Ethics Committee at Curtin University, Perth, Western Australia (HRE2017-0189).

### Data Availability

Czarniak, Petra; Sunderland, Bruce; Parsons, Richard; Lavers, Ainslie; Yip, Wai Siong; Mackenzie, Sarah; Seet, Jason (2018): Data on surgical antibiotic prophylaxis use and infection prevalence in breast surgery procedures at a tertiary hospital in Western Australia. Curtin University. http://dx.doi.org/10.4225/06/5aaf1279876c8

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
