# Peer review of "Surgical antibiotic prophylaxis use and infection prevalence in non-cosmetic breast surgery procedures at a tertiary hospital in Western Australia—a retrospective study"

_PeerJ, doi:10.7717/peerj.5724_

## Round 0.1 · original submission · Major Revisions

Dear authors,

I have received the comments of the reviewers on your manuscript, and copies are included below. The reviewers believe that your manuscript is of potential interest to our readers but feel that substantial revision would be necessary before the paper could be considered again for publication in PeerJ.
Please address the suggestions and concerns raised by the reviewers and submit the revised version.

With respect and warm regards,
Dr Palazón-Bru (academic editor for PeerJ)

Reviewer 1 ·

Basic reporting

1. The use of punctuation marks particularly comma is inadequate. There is need to improve this so as to improve comprehension.
2. Kindly provide a detail legend for Table 3.

Experimental design

1. The title should be reworded. As it is currently captioned, it is not line with the aim and did not reflect the full scope of the study.
2. Evaluating the the impact of prophylactic antibiotics on SSI incidence and length of hospital stay is one of key objective of this study. Unfortunately, there is no table or data in the result section of the manuscript to establish this important relationship. Kindly provide this data and reword your conclusion to show explicit weather there is a relationship between prophylactic antibiotic use and occurrence of SSI.
3. Can you justify the adoption of 250 as appropriate sample size for this study?

Validity of the findings

No comment

Additional comments

this article will make an interesting read for researchers in the field of surgery and infection control practitioners generally. Important areas of concerns have been highlighted in the manuscript.

Annotated reviews are not available for download in order to protect the identity of reviewers who chose to remain anonymous.

Reviewer 2 ·

Basic reporting

Clear professional language used throughout.
Well-structured format.
Tables with clear and relevant information.
Data supplied well.

The current article contains information related to oncologic aspect and management of breast surgery. There is only 1 patient who had Breast Reduction and 1 patient who had implant insertion, in the series of 250. Therefore the article does not cover the entire scope of breast surgeries, with special reference to Aesthetic or Cosmetic aspect of the surgery.
Antibiotics also plays an important role in implant related aesthetic breast augmentation mammoplasty and augmentation mastopexy. Inclusion of the research related to aesthetic aspect of breast surgery and carried out in the past, is important to cover the role of antibiotics related to this spectrum of breast surgery. Without such information, current article will not appear complete or comprehensive. I would strongly recommend the role of duration of antibiotics in breast augmentation mammoplasty contained in the following article:
1. Khan UD. Breast Augmentation, Antibiotic Prophylaxis and Infection: Comparative
Analysis of 1628 Primary Augmentation Mammoplasties to Assess the Role
and Efficacy of Length of Antibiotic Prophylaxis. Aesth Plast Surg. 34:42-47. 2010

Equally there is an article where no antibiotics were given to 899 patiernts who had breast surgery and had a low (2.2%) infection. This reference is also very important to include in this article. (Courtice EH, Goldwyn RM, Anastasi GW (1979) The fate of breast implants with infection around them. Plastic Reconstr Surg 63:812-816

Experimental design

Data ant its analysis is robust, statistically sound.
Experimental Design
Original research.
Research questions well defined and fills an identified knowledge gap.
Methods well defined with reasonable details.

Validity of the findings

Data is robust and statistically sound.
Conclusions are well stated and linked to original research questions.

Additional comments

Congratulation on writing such a great, informative and scientific article highlighting some very important aspects and gapes related to the protocol of Australian Therapeutic Guidelines for the prevention of Surgical Site Infection. However, I would like to recommend changing the title to;
"Surgical antibiotic prophylaxis use and infection prevalence in non-cosmetic breast surgery procedures at a tertiary hospital in Western Australia - a retrospective study".
There are only two procedures in the series of 250, which appears to be aesthetic in nature, These two procedures may well have been performed in patients, admitted for management of breast cancer.

·

Basic reporting

Line 67 what is WA? Please write in full.
Lines 91-94 should be remove or moved to the introductory section.
Lines 128-136 should be remove or moved to the introductory section.
Line 122 to 123 rephrased as follows …………. “were performed using chi square statistics within crosstab menu dialog of spss version ……….”

Experimental design

It is necessary to clarify the type of surgery (clean, contaminated, dirty) etc. Because the rate of infections may differ according this.

What was the criteria used in the determination of SSI from

1. No Clinical correlate between laboratory and infections and antibiotic selection?

2. In order to properly examine the impact of prophylactic antibiotics on SSI incidence and length of hospital stay the author should include data on Microbial pathogens obtained from laboratory investigations the patient's records, Because the rate of infections differ according to this.

3. The author should in addition consider looking at the relationship between SSI, type of surgery and microbial distribution and include (if available) record the hand wash before approaching the patient.

Validity of the findings

The scientific level is not very high appears to be only data collection.
Why chi-square test and not ANOVA or MANOVA?

As a limitation, the author should indicate that no post-discharge infections were recorded for this study.

Finding are valid

Conclusions needs will need slight modification if corrections are made.

Additional comments

A well written manuscript. However the suggested modifications should be made.

---

## Round 0.2 · Minor Revisions

Dear authors,

Still pending a minor correction in your paper before being considered for publication in PeerJ. Please, see the comments of the reviewers in order to have more information.

With respect and warm regards,
Dr Palazón-Bru (academic editor for PeerJ)

Reviewer 1 ·

Basic reporting

The authors have satisfactorily responded to the comments. However, the authors have earlier published an article in 2017 with a similar aim as the current one ''To examine adherence with therapeutic guidelines (eTG) in breast surgery .. ''. May the authors justify the need for the current publication and assure the journal that this is not is not a duplicate publication.
As stated in the last review, discussion is not necessary in abstract. The authors should preferably state their conclusion and possible implications of their findings.

Experimental design

Satisfactory

Validity of the findings

The authors have conducted an excellent review of the level of adherence to therapeutic guidelines, well written with professional language.

Reviewer 2 ·

Basic reporting

Clear professional language used through out.

Experimental design

The article contains information related to the oncologic aspect of breast surgery and this original research clearly meets aim and scope the Journal.

Validity of the findings

I am happy with the experimental design of the study. The data presented is robust and statistically sound.

Additional comments

Thanks for making the changes to the article. The article itself contain useful information based on scientific research and data collection.

·

Basic reporting

I find it satisfactory. the authors has address all my observations

Experimental design

Adequate,considering his stated limitations.

Validity of the findings

A good work.

Additional comments

The author addressed all off my comments humbly.

---

## Round 0.3 · accepted · Accept

Dear authors,

I am happy to inform you that your paper has been accepted for publication in its current form in PeerJ.

Congratulations!

With respect and warm regards,
Dr Palazón-Bru (academic editor for PeerJ)

Reviewer 1 ·

Basic reporting

My concerns have been addressed.

Experimental design

Satisfactory

Validity of the findings

Excellent